# Use of "The Knowledge-to-Action Framework" for the implementation of evidence-based nursing in child and family care: Study protocol

Cânia P. Torres[1,2,3,4]*, Francisco J. Mendes[2,3], Maria Barbieri-Figueiredo[1,2,5]

1 Institute of Biomedical Sciences Abel Salazar, University of Porto, Porto, Portugal, 2 Center for Health Technology and Services Research at the Health Research Network (CINTESIS@RISE), Porto, Portugal, 3 Nursing Department, Pediatrics Service, São João Hospital Centre, Porto, Portugal, 4 Nursing School of Tâmega e Sousa, CESPU, Penafiel, Portugal, 5 Nursing Department, University of Huelva, Huelva, Spain

* caniabasto@gmail.com

## Abstract

Nurses are increasingly focused on a practice based on scientific knowledge. However, it is important to distinguish high-quality evidence that can be applied in practice from studies of low or dubious scientific quality. Therefore, nurses must base their practice on structural support that allows for the definition of personalized and context-specific interventions. The objectives of this study are to identify the main barriers and facilitators to the evidence-based nursing approach and to implement an Evidence-Based Practice model (EBP) in clinical practice settings. We seek to contribute to evidence-based nursing by promoting professional skills in nurses, using "The Knowledge-to-Action Framework" (KTA). The research focuses on a participatory action research methodology based on the cyclical process of the KTA framework, contemplating the creation of knowledge and the implementation of existing solutions or new solutions through an action cycle. The participants will be nurses and parents/caregivers) from a pediatric service in Northern Portugal. The study will be conducted in 3 phases: phase 1 will identify the priority issues by exploring the barriers and facilitators of EBP from the nurses' perspective and assessing the parents'/caregivers' satisfaction with nursing care. Phase 2 will be divided into (a) the planning and (b) the implementation of the KTA model, where we aim to build and validate (a) a training plan and (b) the implementation of the KTA model. Phase 3 is for the evaluation of the model implementation and sustaining knowledge. It is recognized that there is a large gap between knowledge production and the subsequent implementation of interventions based on the best available evidence. However, this reality is complex and involves several levels of decision and intervention that oscillate from the individual responsibility of each nurse to the organizational dimension.

**Funding:** The author(s) received no specific funding for this work.

**Competing interests:** The authors have declared that no competing interests exist.

# Background

The EBP movement, which was strongly propelled in the 1990s, raised awareness of the importance of health professionals' clinical decisions being informed by the best available evidence while considering their clinical experience as professionals, people's beliefs, values and concerns, and the clinical practice setting in which the care is delivered [1]. Adopting specific Evidence-Based Nursing (EBN) competencies for nurses and nurse specialists practicing in complex healthcare settings can help institutions achieve higher-value, lower-cost evidence-based healthcare. However, it is still not the standard that is seen in the care provided by health professionals [2]. EBP is a lifelong problem-solving approach to health care delivery that integrates the best evidence from well-designed studies, such as external evidence, with each patient's preferences and values. When EBP is part of the care delivery context, as safety culture, the best clinical decisions are taken, generating positive outcomes for the nurse, the patient, and the family [2, 3]. Thus, the empowerment of nurses becomes critical to support their decision-making [4].

Professional empowerment of nurses is related to organizational outcomes, so it is important to strengthen and optimize organizational structures [4, 5]. Empowerment can be achieved with access to structures that promote empowerment, particularly resources, where investment in training is expected to have a very positive impact on research, teaching, and health care delivery, promoting the development of systematic reviews and other quality studies and consequently allowing the best scientific knowledge to be updated and integrated, resulting in a positive impact on health care and health education [3–5]. It is widely accepted that global healthcare policies, programs, and practices need to be based on the best available scientific evidence. However, in practice, it has been difficult to obtain scientific knowledge that is drawn from the implementation of models of EBP in all settings, and it is here that we find large gaps in knowledge about how to overcome this problem [6].

The implementation of models for EBP in hospital nursing aims to promote improved quality of care by increasing the reliability of interventions. The models describe steps that range from research and selection of the best evidence to strategies to ensure the sustainability of its incorporation in hospital organizations [7]. The implementation of the models requires nurses to know methodological approaches and different types of research, critical analysis of publications (primary studies), as well as methods of synthesizing the results of primary studies (e.g., systematic review) [6, 7]. Given the practical context where this study will be developed, we have chosen to use the Knowledge to Action (KTA) framework, which was originally developed in 2006 by Dr. Ian Graham and colleagues and stems from the identification of several key steps in the complex process of translating knowledge into action, resulting from the review and aggregation of over 60 theories, frameworks, and action planning models. This framework provides a structured approach to making change, including a six-stage action cycle that enables the translation of knowledge into practice [8].

Monitoring, evaluating, and disseminating the results are crucial components for using research in nurses' practice. Thus, by bringing together the stakeholders, nurses, and parents, we will understand the problems in practice and, through the implementation of the KTA framework, implement an EBP in the care provided to the child and family. Accordingly, this article describes a protocol that allows for the implementation of EBP through an EBN program designed to develop mentors in professional and academic settings, with the potential to transform a healthcare organization and towards a logic of change and improvement of nursing care [6, 9]. The following objectives will guide the study: (1) to identify the priority issues by exploring the barriers and facilitators of EBP from the nurses' perspective and assessing the parents'/caregivers' satisfaction with nursing care. (2) build and validate (a) a training plan

and (b) the implementation of the KTA model. (3) evaluation of the model implementation and sustaining knowledge.

## Pertinence of the study

With this protocol it is intended to build a conceptual and methodological basis for the study, based on the KTA model. The translation of knowledge in nursing practice is currently a focus for professional practice in family nursing and several studies evidence that this practice has very significant contributions in improving care [10]. The inclusion of stakeholders in all crucial stages of the research process will guarantee that it will be of interest to all the providers of nursing care to the child and family. With the involvement of pediatric nurses and parents/caregivers of hospitalized children, it is expected to evaluate the real contribution that the implementation of KTA model will have in care, from the perspective of all involved. Although in Portugal there is no study in the literature that uses KTA model in this context, in other countries the reality is different. This study provides nurses with a concrete strategy that allows structuring the process of professional practice of nurses, as regulated in the common competencies of nurses in the light of the National Nursing Association (Ordem dos Enfermeiros).

## Structure of the Knowledge-to-Action Framework

It is important to know that the knowledge creation part of the framework depicts processes used to identify relevant knowledge and validate and adapt it to the specific area of knowledge search. This is the process used in developing evidence-based guidelines. Research and other evidence are identified and synthesized into knowledge tools and products, such as practice recommendations, clinical pathways, and patient decision support. Essentially, all guideline development methodologies incorporate a knowledge-creation process, although some are more rigorous than others [11]. The action cycle is the process by which the knowledge created is implemented, evaluated, and sustained in the clinical setting.

The six stages that make up the action cycle are based on a synthesis of evidence-based theories that focus on the process of intentional and systematic change in healthcare organizations [11, 12]. The first stage of the action cycle–identifying the problem–is operationalized in two ways [11, 12]: (1) Nurses define a problem (usually through quality improvement methods and/or data analysis) and then identify possible recommended practices that can help solve the problem. (2) Nurses learn about a good practice guide (GPG)/protocol and determine if current practice is meeting the evidence or if practice change is needed. This initial phase is important because it defines the level of stakeholder participation in the implementation process, how it is related to continuous quality improvement, and what the identified problem will look like. The option should start with adopting best practices, education, and/or policy development or revision [12].

The second stage–adaptation to the local context–is key to effective knowledge translation and requires understanding the local context and the implication of best practices in that context. Adapting a guideline to the local context does not disregard the evidence base that supports the guideline. Adaptation involves identifying the stakeholders responsible for implementing, reviewing, and selecting appropriate guidelines for use, evaluating them for quality, and determining their clinical utility and suitability to be implemented. Modifications to the guideline recommendations can be made at this point by the implementation group with stakeholders. The goal of this phase is to select a knowledge product in a transparent way that is perceived by all to reflect the best evidence about the identified problem and to fit the local context. This phase involves participants and stakeholders at a planning

level to ensure that the selected GPG will meet the needs and can be tailored to the organizational context [11, 12].

The third stage–assessment of facilitators and barriers to using knowledge–identifies barriers and facilitators to using best practice guidelines (BPG) in the clinical setting. Specific barriers, such as lack of knowledge, attitudes, and resistance to change, are points to retain for those elements that will facilitate knowledge transfer. At this stage, it is also important to identify the stakeholders who support the selected BPG implementation work. In essence, this phase incorporates an assessment of the clinical setting and relevant stakeholders [12, 13].

The fourth stage–adaptation and implementation of interventions–incorporates an implementation plan that considers stakeholder assessment and implementation strategies. It also includes an assessment of barriers and facilitators and evidence on effective implementation strategies. The plan builds on this data to identify and support the selection of interventions that will facilitate the implementation of new knowledge in the clinical setting [12, 13].

The fifth and sixth stages concern monitoring, evaluation, and sustaining the use of knowledge and are central to implementation. These phases include assessing the use of BPG as demonstrated by adherence to recommendations or changes in knowledge and/or attitudes, evaluating the impacts or outcomes of implementing the BGP recommendations and sustaining the changed practices. Given the importance of these aspects of the knowledge-to-action cycle, they must be considered in all the previous phases. Similarly, careful planning and implementation of the first four phases provide direction for the monitoring, evaluation, and incorporation processes [12, 13].

## Methods

This study will follow a participatory action research (PAR) approach to health, focusing on a continuum between qualitative and quantitative: multi-methods [14]. The methodological strategy PAR is increasingly used because it aims to identify problems and their solutions [15], which makes it a favorable and useful element in Nursing research. Nursing seeks to expand the theoretical body of the profession by conducting both quantitative and qualitative research, made possible by methodological strategies that best suit the type of research. The PAR, used when there is a collective interest in solving problems, is among these strategies [15, 16]. It should be understood as a means of empowerment, that is, as transforming participation that implicitly should lead to all people being able to carry out processes of awareness [15, 16], resulting in reflections, predisposition to act, and changing behaviors.

According to the International Collaboration for Participatory Health Research (ICPHR), the PAR is an approach that combines qualitative and quantitative research methods and techniques, where the participation of the target group is central. It considers processes, results, and impact [17]. The study will be carried out in 3 phases to meet the proposed objectives and respect the steps of the KTA framework [8]. At the core of the whole development of the study is the knowledge creation process, also identified as the "knowledge funnel" [13]. This process is the basis for creating a knowledge product/tool, such as a best practice guideline, i.e., a decision aid. It is at the center of the KTA diagram. It looks like a funnel because it gathers all the available knowledge on a topic, summarizes the inside, and creates a tool that can be used to improve care [17]. The action cycle includes 6 stages and does not necessarily have to be completed sequentially. Still, all will be addressed in this study, and, as envisaged in the KTA framework, we plan to work on several stages of this cycle simultaneously [8, 18].

This study was approved by the Ethics Committee for Health and the Research Unit of the Institution where it was held (approval number 93/2021).

## Study sample

The study participants will be selected from a pediatric service of a hospital in the north of Portugal. Therefore, nurses and parents/caregivers of hospitalized children are the target population of our study so that we can understand the contribution that the implementation of this evidence-based practice model has on the nursing care provided to children and their families. It is essential to study both sides of care: the providers (nurses) and the receivers (caregivers/parents of hospitalized children). In phase 1 of the project, we aim to recruit all the nurses working in the service where the study is being developed, i.e., n = 32. The participants will always be recruited in the same service. The nurses will be selected intentionally since the goal is that the whole nursing team of the pediatric service is involved in the study. The parents/caregivers of the hospitalized children will be chosen randomly, following one criterion, i.e., that their children had been hospitalized for more than 24 hours in the pediatric ward. Participants for phase 2 will now be recruited from this initial sample, i.e., n = 32 nurses, i.e., the same as those who participated in phase 1 and involving caregivers of hospitalized children, at least 100 participants. In phase 3, participants will be recruited from the phase 1 sample, hopefully the total of nursing team, i.e. 32 nurses and predictably at least 100 caregivers of children admitted to the service at the time this phase of the study takes place.

It should be noted that we considered that some sample could happen and, to safeguard them, we indicate some considerations. All nurses who are working in the service where the study will take place will be eligible to participate, however, the nurses participating in Phase I of the study will be the same as those who will be able to integrate Phases II and III. If new nurses are integrated into the service during the process, they will not be considered for the sample, but may, in the future and after the end of the study, be integrated. Nurses who, maintaining a bond with the institution, are absent from the service in phases II and/or III for personal reasons, may continue to participate in the study if they wish so.

## Research design

**Phase 1: Identifying the problem.** In this first phase of the study, we aim to identify barriers and facilitators to EBP and explore nurses' opinions regarding their professional practice setting. The stages of the KTA framework included in this study are three, and they have distinct objectives [13]: (a) Identify, review, and select Knowledge: Determining the knowledge gap in clinical practice; (b) Identify stakeholders in the clinical context: nurses and children's caregivers; (c) Assess Barriers / Facilitators to Knowledge use: Assess barriers and facilitators to knowledge.

(Stage 1) Identify gaps in knowledge for intervention: in this stage, the framework aims to identify gaps in knowledge to intervene, which implies assessing the knowledge needs and is considered the starting point of the whole process. The use of methods and instruments must be rigorous and consider the different perspectives (of caregivers/parents, nurses, and the organization) [22, 23]. In this first stage, the 32 nurses recruited for the study will be surveyed using the Questionnaire of Attitudes and Barriers to Evidence-Based Practice (QABPBE-26) [19], an instrument to be filled out individually to identify barriers and facilitators to EBP in the specific scenario where the study will be developed. Regarding the caregivers of hospitalized children, the following instruments will be applied: the Family Support Perception Questionnaire (QPSF) [20] and the Citizen Satisfaction with Nursing

Care Scale for parents of hospitalized children (ESCCE) [21]. (Stage 2) Adapting knowledge to the local context and (Stage 3) Assess barriers to the use of knowledge: The evidence to support the practice should be adapted to the context for which it is intended so that it can contribute to better acceptance and adherence to this practice, overcoming various obstacles such

as the lack of skills in EBP and expertise of professionals. The assessment of the barriers to using knowledge takes on special importance and may be related to different issues, ranging from behavioral, attitude, and motivation to external factors [22]. Using a semi-structured interview script and the focus group methodology, we will divide the nurses participating in the study into groups to explore their opinions regarding the problems in their professional practice. Thus, we will be able to adapt knowledge to the context better and work on existing barriers.

The questionnaires that are expected to be used at this stage of the study all have psychometric validation for the Portuguese population, thus guaranteeing validity and reliability. The QABPBE-26 Portuguese version presents 26 items with an acceptable internal consistency (Cronbach's alpha = 0.60). The Main Component Analysis suggests eight dimensions that explain 55.77% of total variance. The analysis conducted demonstrated valid empiric evidence and the questionnaire could be used in our context [19]. The QPSF presents an internal consistency of 0.94 [20]. The ESCCE instrument for parents of hospitalized children [21], has adequate psychometric characteristics for the Portuguese population of parents of hospitalized children (Cronbach's alpha = 0.92), ensuring its reliability and validity for measuring satisfaction with nursing care.

**Phase 2: Planning and implementing the framework.** A participatory action research study, where the main objective is to promote the development of nurses' skills for EBP in child and family care. The stage of the KTA model included in this phase of the research protocol has the following objectives (12): (d) Establish a robust infrastructure: nurses' empowerment—training; (e) Select, Tailor, and Implement Interventions/Implementation Strategies: Implementation strategies.

(Stage 4) Select adapted interventions for implementation: Based on the results obtained in phase 1, with the instruments applied to nurses and children's caregivers, we will build and validate a training plan for nurses. A transfer of knowledge implies an adaptation of the interventions to overcome barriers that may hinder the process [22, 23]. It will be important to use different educational interventions aimed at practical scenarios and nurses' decision-making, directed and centered on children and parents/caregivers. Institutional and organizational interventions should not be excluded [25]. The construction of the training plan with contents related to EBP, based on a literature review and its subsequent validation, will be carried out using the Delphi methodology. The experts who will integrate this validation will be nurses/teachers/researchers with scientific publications and work developed in EBP. In this phase, the training plan, distributed by modules, will also be implemented. The training will be divided into basic training, aimed at all the nurses who participate in the study, and advanced training, as a complement for the nurses who will be designated as mentors. The mentors will be chosen from among specialist pediatric nurses who, from the perspective of the head nurse of the service, are facilitators of change and capable of sustaining it over time. Also, in this phase, and after the nurses' training process is completed, lines of research in clinical practice will be implemented, which emerged from phase 1 of the study and address the main issues identified by the stakeholders (caregivers and nurses).

**Phase 3: Evaluation of the framework implementation.** The main objectives set out in the framework for the stages included in this research phase are (f) Monitor Knowledge Use & Evaluate Outcomes: Identify key indicators; Concepts of knowledge; (g) Implementation of results; (h) Strategies to sustain. In this last phase of the research, we aim to measure the impact of knowledge use on parents/caregivers and nurses, considering the results obtained. The impact of using knowledge should be measured considering the results obtained with patients, professionals, and the organization. The sustainable use of knowledge is the way forward, along with its evaluation and monitoring [2]. The steps included in this last phase of the research are steps 5 and 6, which are explained below.

(Stage 5) Monitoring the use of knowledge and evaluating gains: Knowledge usage will be monitored conceptually and strategically and assessed with the stakeholders (nurses and parents/caregivers) [23]. The data collection instruments we will use will be the same as those used in Phase 1 because we hope to be able to measure changes that come from implementing the KTA framework [20, 21]. The evaluation of the impact of the EBP training process on nurses will also be assessed through the QABPBE-26 questionnaire and the involvement of nurses and co-researchers in the research development and evidence implementation processes. (Stage 6) Sustainable Use of Knowledge: Sustainability, in global terms, derives from the set of factors addressed in the previous stages. Integrating mentors, co-researchers, and a nurse expert from academia in each research line will help sustain the KTA framework in the practice of nurses. Each research line will follow the KTA to implement new knowledge, following the stages that compose it.

Fig 1 below shows the 3 stages of the Research Protocol: *Use of the Knowledge-to-Action Framework for the Implementation of Evidence-Based Practice in Child and Family Care*, the six (6) phases of the KTA framework action cycle. Each of the project stages contributes to the knowledge-creation process. At the center of the framework is the "knowledge funnel", composed of three stages: Questioning knowledge, knowledge synthesis, and knowledge products and instruments.

## Creation of knowledge

Focusing on what is common to all stages of the KTA framework and phases of our study protocol, i.e., the knowledge cycle/funnel—from knowledge to practice, we analyze the 3 levels that constitute this cycle, and that underpin the Cycle of Action presented above [13, 21, 22]: Questioning knowledge (1); Knowledge synthesis (2) and Knowledge products and instruments (3). Questioning knowledge (1) refers to "primary knowledge", where the supply is immense and often unrefined, and access is mostly through digital platforms. At this stage, we refer to the synthesis of knowledge, usually recognized as "second-generation knowledge", where existing knowledge is aggregated, promoting the synthesis of studies that aim to answer specific questions (2). At the third level of the Knowledge Funnel are the knowledge products and tools associated with the "third generation of knowledge", composed of clearly presented synopses of knowledge. They constitute explicit recommendations that fulfill the needs of the various stakeholders, influencing their actions. Examples of these decision support tools are the Clinical Guidelines [23, 25]. The study protocol follows the research design presented in Fig 2, where 4 studies are planned and divided into 3 phases, with the participatory action research methodology present in all the phases.

## Discussion

In the initial phase, this protocol aims to investigate/study nurses' attitudes and barriers to evidence-based nursing and the use of research knowledge in professional practice (Phase 1). After identifying the problems, we aim to empower nurses in EBP by implementing a training program planned and validated according to the problems identified by our stakeholders (Phase 2). Using the KTA framework, we aim to implement EBP in nursing care provided to children and their families. The creation of knowledge and its translation into practice can support and sustain nurses' decision-making (Phase 3). It seems relevant to us the built figure of the mentors in EBP because they will be important links to the dissemination and maintenance of knowledge. We understand that there is a growing concern for EBP due to its relationship with quality improvement in care and potential cost reduction. However, we expect some limitations to the study. As with any research based on the participatory action research

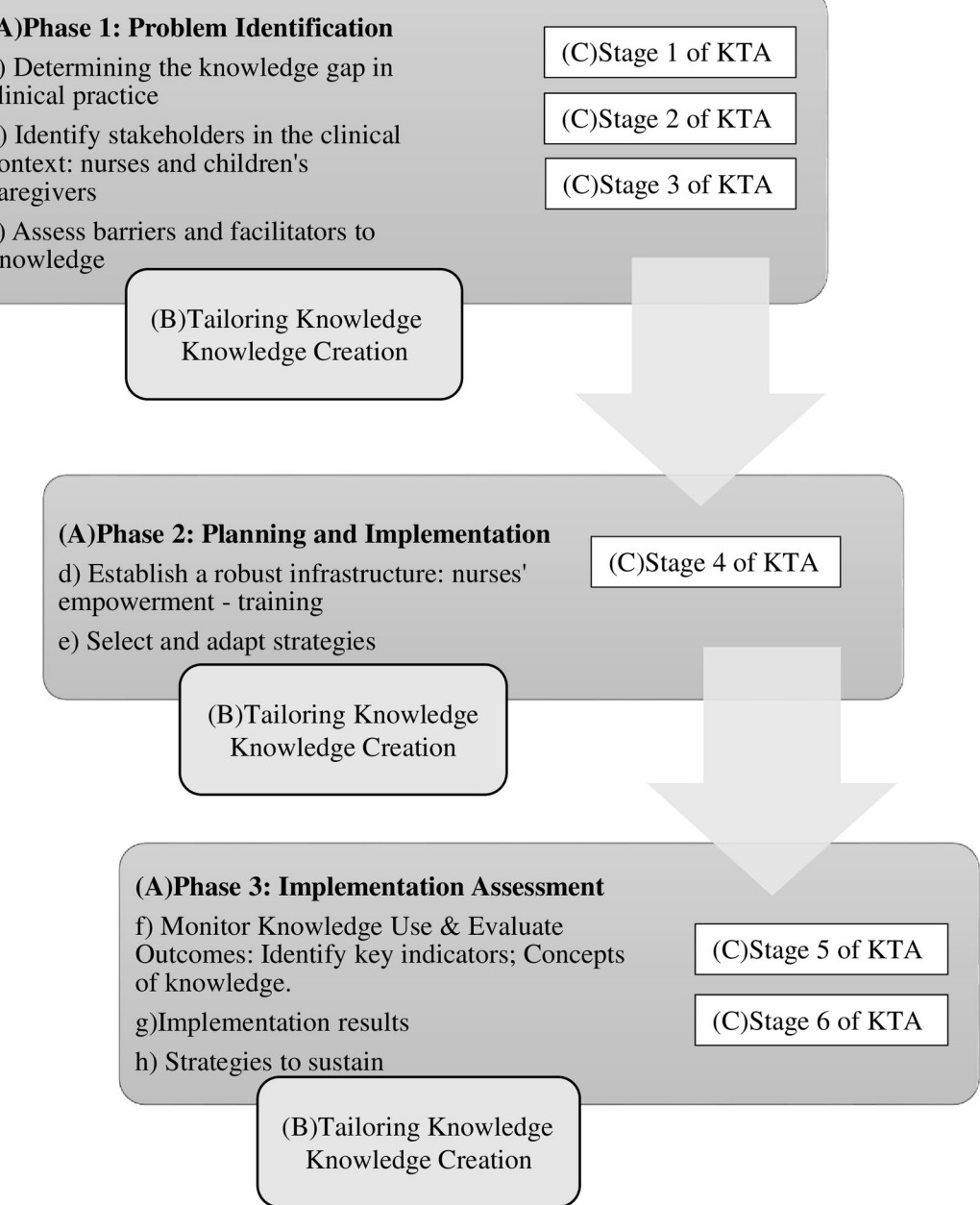

**Fig 1. KTA framework and structure of the research project** [8, 12, 13]. (A) Phases of research. (B) Creation of Knowledge, KTA. (C) Stages of the Action Cycle, KTA.

methodology, we always depend on the participants' interest/acceptance of being included in the study. The researchers' recognition that parents/caregivers and nurses are the stakeholders of the research, assuming that nurses are co-researchers, further links success to their participation. These possible limitations do not diminish the relevance of the proposed protocol, particularly considering that this practice has the potential to provide safe practice environments for patients and healthcare professionals. It is expected frequent meetings to occur during design and implementation phases, establishing trust among the researchers and hospital system leaders to support communication.

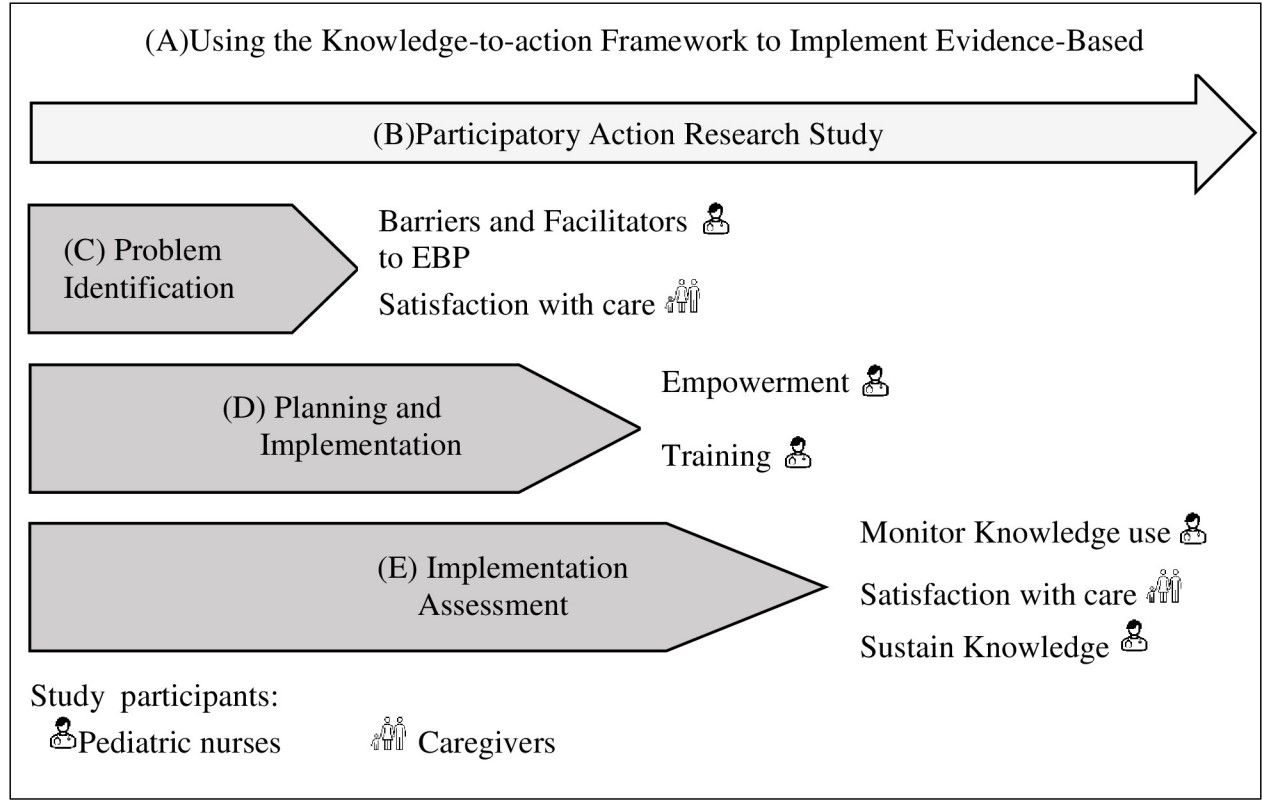

**Fig 2. Research design** [13, 23, 24]. (A) Research topic. (B) Methodology. (C) Phase 1 of the research. (D) Phase 2 of the research. (E) Phase 3 of the research. (F) Study I, Phase 1 of the research. (G)Study II, Phase 1 of the research. (H) Study III, Phase 2 of the research. (I) Study IV, Phase 3 of the research.

## Study contributions

This study may bring important contributions to the practice of nurses, health organizations and future studies. Its relevance in innovation and continuous improvement of pediatric nursing care is considerable because pediatric nurses are sought to integrate evidence into their professional practice. With the empowerment of nurses, through the training and monitoring of mentors in EBP, it will allow to produce evidence directed to the real context. In the literature, it is perceived that the use of the best evidence in the practice of nurses is a challenge, and, in this project, we intend to develop strategies that potentiate the implementation of this evidence. Through the KTA model, and its structure, it will be possible to develop and sustain an environment that facilitates the translation of knowledge, specifically directed to the reality experienced, involving stakeholders: nurses, parents/caregivers in the crucial phases of the study. Also implies the development of nursing resources in the institution. With the EBP training that this study provides to nurses, it will allow them to develop competencies in clinical research and health services. Thus, the health organization will have more qualified human capital to carry out research. At the level of clinical governance there will be an impact, because it promotes the reduction of the variation of the practice of nurses and consequent increase in effectiveness. This study seems to offer support for the development of future research related to the use of KTA model or other, for the production, synthesis, and implementation of evidence in the practice of hospital nurses, seeking to contribute to the increase of scientific production in this theme, still challenging for nursing in Portugal.

## Conclusion

The implementation of KTA model in a pediatric sane hospital unit will provide valuable strategies that will allow the implementation of evidence-based nursing in the care provided to children and families. It will allow nurses to add scientific knowledge to their skills in caring for sick children and parents/caregivers, involving the family and their satisfaction with nursing care in this process. The assessment of the satisfaction of parents/caregivers before and after the implementation of the KTA model will be extremely relevant and will be fundamental for the definition of existing problems. In this study, the involvement of stakeholders, becoming partners in this process, has the potential to motivate the very conduct of the study, since it is expected a very positive influence, both in the development of competencies in EBP in nurses, and the improvement of the satisfaction of parents/caregivers with the care provided in the hospital unit.

We hope with the implementation of KTA model, positively influence the view of pediatric nurses, on the use of scientific evidence in care. The implementation of this study will effectively be a challenge with an impact in pediatric nursing practice.

## Author Contributions

**Conceptualization:** Cânia P. Torres.

**Data curation:** Cânia P. Torres.

**Formal analysis:** Cânia P. Torres.

**Investigation:** Cânia P. Torres.

**Methodology:** Cânia P. Torres, Francisco J. Mendes, Maria Barbieri-Figueiredo.

**Project administration:** Cânia P. Torres.

**Resources:** Cânia P. Torres.

**Supervision:** Cânia P. Torres, Francisco J. Mendes, Maria Barbieri-Figueiredo.

**Validation:** Cânia P. Torres.

**Visualization:** Cânia P. Torres.

**Writing – original draft:** Cânia P. Torres, Francisco J. Mendes, Maria Barbieri-Figueiredo.

**Writing – review & editing:** Cânia P. Torres, Francisco J. Mendes, Maria Barbieri-Figueiredo.

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
