## [Decision Letter · Decision Letter 0]

1 Feb 2023

PONE-D-22-33958Uso do “The Knowledge-to-Action Framework” para a implementação da enfermagem baseada em evidências no cuidado à criança e à família: protocolo de estudoPLOS ONE

Dear Dr. Torres,

Thank you for submitting your manuscript to PLOS ONE. After careful consideration, we feel that it has merit but does not fully meet PLOS ONE’s publication criteria as it currently stands. Therefore, we invite you to submit a revised version of the manuscript that addresses the points raised during the review process.

We look forward to receiving your revised manuscript.

Kind regards,

Majed Sulaiman Alamri, PhD

Academic Editor

PLOS ONE

Journal Requirements:

2.During your revisions, please confirm whether the wording in the title is correct and update it in the manuscript file and online submission information if needed. Specifically, the title requires translation into the English language.

Reviewers' comments:

Reviewer's Responses to Questions

**Comments to the Author**

1. Does the manuscript provide a valid rationale for the proposed study, with clearly identified and justified research questions?

Reviewer #1: Yes

Reviewer #2: Yes

2. Is the protocol technically sound and planned in a manner that will lead to a meaningful outcome and allow testing the stated hypotheses?

Reviewer #1: Yes

Reviewer #2: Yes

3. Is the methodology feasible and described in sufficient detail to allow the work to be replicable?

Reviewer #1: Yes

Reviewer #2: Yes

4. Have the authors described where all data underlying the findings will be made available when the study is complete?

Reviewer #1: No

Reviewer #2: No

5. Is the manuscript presented in an intelligible fashion and written in standard English?

Reviewer #1: Yes

Reviewer #2: Yes

6. Review Comments to the Author

You may also provide optional suggestions and comments to authors that they might find helpful in planning their study.

Reviewer #1: The topic of this manuscript, the Knowledge To Action Framework, is of great value to nurses and other readers interested in translation science.

For that reason, I believe that the authors of this manuscript is too ambitious in presenting the KTA Framework as well as the protocol for a future study for which the model is foundational. I believe this manuscript needs to be a stand-alone review paper which describes in much greater detail the process of developing the KTA Framework: What theoretical gap did this KTA Framework fill, What databases and parameters were used for the search, what inclusion criteria were applied to the theories / models that were examined, what are the elements of the KTA Framework, how may the framework be applied in translation science, and how may it be evaluated?

This will give you a solid theoretical foundation when you proceed to report the findings of your proposed study, and I believe will foster greater dissemination of the framework than the combined manuscript will.

Reviewer #2: I was very interested and enjoyed while I am reading your research protocol. However, there were some points need to be considered if you would like to publish this peace of paper.

1. Background: Although your explanation about the structure of Knowledge-to-Action framework principles and dimensions is great, it would be lovely to add paragraph explaining the rational for your study and why other literature did not cover this area.

2. Sample: Please write more details about recruitment process and how you avoided bias while collecting your data.

3. Data collection: In research design section (under phase 1), you mentioned some research instruments, but you haven’t explained if they are valid or psychometrically evaluated. Please add a validity and reliability score for each.

4. Discussion: You need to add a section containing the expected contribution of the study to practice, policy and future research.

5. You need also to add conclusion at the end addressing all key findings together in a good manner.

7. PLOS authors have the option to publish the peer review history of their article (what does this mean?). If published, this will include your full peer review and any attached files.

Reviewer #1: No

Reviewer #2: No

---

## [Author Response · Author response to Decision Letter 0]

14 Feb 2023

Cânia Torres

e-mail: caniabasto@gmail.com

 Editor-in-Chef

 PLOS ONE

Dear Editor-in-Chef,

Please find attached the revised version of the manuscript entitled “Use of The Knowledge-to-Action Framework for the implementation of evidence-based nursing in child and family care: study protocol”. All modifications were made in accordance to reviewers’ considerations and the answers to their comments are also attached. In our opinion significant modifications were carried out across the manuscript. The entire manuscript was revised by an English native speaker.

Thanks for your attention.

Yours sincerely,

Cânia Torres

Answer to Reviewer 1

We would like to thank for the reviewer effort and dedicated time to evaluate our manuscript.

Point 1:

The reviewer suggested “For that reason, I believe that the authors of this manuscript is too ambitious in presenting the KTA Framework as well as the protocol for a future study for which the model is foundational. I believe this manuscript needs to be a stand-alone review paper which describes in much greater detail the process of developing the KTA Framework. (…)”

Response 1: We are grateful for the suggestion, but carrying out an isolated review is not foreseen within the scope of the study protocol.

Answer to Reviewer 2

We would like to thank the reviewer effort and dedicated time to evaluate our manuscript. The points addressed allowed to improve considerably our work.

Point 1:

Background: Although your explanation about the structure of Knowledge-to-Action framework principles and dimensions is great, it would be lovely to add paragraph explaining the rational for your study and why other literature did not cover this area.

Response 1: A new point has been introduced in the document: Pertinence of the study.

“With this protocol it is intended to build a conceptual and methodological basis for the study, based on the KTA model. The translation of knowledge in nursing practice is currently a focus for professional practice in family nursing and several studies evidence that this practice has very significant contributions in improving care. (25,26) The inclusion of stakeholders in all crucial stages of the research process will guarantee that it will be of interest to all the providers of nursing care to the child and family. With the involvement of pediatric nurses and parents/caregivers of hospitalized children, it is expected to evaluate the real contribution that the implementation of KTA model will have in care, from the perspective of all involved. Although in Portugal there is no study in the literature that uses KTA model in this context, in other countries the reality is different. This study provides nurses with a concrete strategy that allows structuring the process of professional practice of nurses, as regulated in the common competencies of nurses in the light of the National Nursing Association (Ordem dos Enfermeiros)” (line 126 to 140).

Point 2:

Sample: Please write more details about recruitment process and how you avoided bias while collecting your data.

Response 2: At the point of the Study Sample, it was added “It should be noted that we considered that some sample could happen and, to safeguard them, we indicate some considerations. All nurses who are working in the service where the study will take place will be eligible to participate, however, the nurses participating in Phase I of the study will be the same as those who will be able to integrate Phases II and III. If new nurses are integrated into the service during the process, they will not be considered for the sample, but may, in the future and after the end of the study, be integrated. Nurses who, maintaining a bond with the institution, are absent from the service in phases II and/or III for personal reasons, may continue to participate in the study if they wish so” (line 254 to 261). 

Point 3:

Data collection: In research design section (under phase 1), you mentioned some research instruments, but you haven’t explained if they are valid or psychometrically evaluated. Please add a validity and reliability score for each.

Response 3: At the point of the Research Design - Phase 1: Identifying the Problem, “The questionnaires that are expected to be used at this stage of the study all have psychometric validation for the Portuguese population, thus guaranteeing validity and reliability. The QABPBE-26 Portuguese version presents 26 items with an acceptable internal consistency (Cronbach's alpha =0.60). The Main Component Analysis suggests eight dimensions that explain 55.77% of total variance. The analysis conducted demonstrated valid empiric evidence and the questionnaire could be used in our context (18). The QPSF presents an internal consistency of 0.94 (19). The ESCCE instrument for parents of hospitalized children (20), has adequate psychometric characteristics for the Portuguese population of parents of hospitalized children (Cronbach's alpha = 0.92), ensuring its reliability and validity for measuring satisfaction with nursing care" (line 296 to 305). 

Point 4:

Discussion: You need to add a section containing the expected contribution of the study to practice, policy and future research.

Response 4: A point was added that refers to the contributions of the study. “This study may bring important contributions to the practice of nurses, health organizations and future studies. Its relevance in innovation and continuous improvement of pediatric nursing care is considerable because pediatric nurses are sought to integrate evidence into their professional practice. With the empowerment of nurses, through the training and monitoring of mentors in EBP, it will allow to produce evidence directed to the real context. In the literature, it is perceived that the use of the best evidence in the practice of nurses is a challenge, and, in this project, we intend to develop strategies that potentiate the implementation of this evidence. Through the KTA model, and its structure, it will be possible to develop and sustain an environment that facilitates the translation of knowledge, specifically directed to the reality experienced, involving stakeholders: nurses, parents/caregivers in the crucial phases of the study. Also implies the development of nursing resources in the institution. With the EBP training that this study provides to nurses, it will allow them to develop competencies in clinical research and health services. Thus, the health organization will have more qualified human capital to carry out research. At the level of clinical governance there will be an impact, because it promotes the reduction of the variation of the practice of nurses and consequent increase in effectiveness. This study seems to offer support for the development of future research related to the use of KTA model or other, for the production, synthesis, and implementation of evidence in the practice of hospital nurses, seeking to contribute to the increase of scientific production in this theme, still challenging for nursing in Portugal" (line 421 to 442).

Point 5:

You need also to add conclusion at the end addressing all key findings together in a good manner.

Response 5: A conclusion was included in the document. “The implementation of KTA model in a pediatric sane hospital unit will provide valuable strategies that will allow the implementation of evidence-based nursing in the care provided to children and families. It will allow nurses to add scientific knowledge to their skills in caring for sick children and parents/caregivers, involving the family and their satisfaction with nursing care in this process. The assessment of the satisfaction of parents/caregivers before and after the implementation of the KTA model will be extremely relevant and will be fundamental for the definition of existing problems. In this study, the involvement of stakeholders, becoming partners in this process, has the potential to motivate the very conduct of the study, since it is expected a very positive influence, both in the development of competencies in EBP in nurses, and the improvement of the satisfaction of parents/caregivers with the care provided in the hospital unit. We hope with the implementation of KTA model, positively influence the view of pediatric nurses, on the use of scientific evidence in care. The implementation of this study will effectively be a challenge with an impact in pediatric nursing practice” (445 to 460).

---

## [Decision Letter · Decision Letter 1]

14 Mar 2023

Use of “The Knowledge-to-Action Framework” for the implementation of evidence-based nursing in child and family care: study protocol

PONE-D-22-33958R1

Dear Dr. Torres,

We’re pleased to inform you that your manuscript has been judged scientifically suitable for publication and will be formally accepted for publication once it meets all outstanding technical requirements.

Kind regards,

Majed Sulaiman Alamri, PhD

Academic Editor

PLOS ONE

Additional Editor Comments (optional):

Reviewers' comments:

Reviewer's Responses to Questions

**Comments to the Author**

1. Does the manuscript provide a valid rationale for the proposed study, with clearly identified and justified research questions?

Reviewer #1: Yes

Reviewer #2: Yes

2. Is the protocol technically sound and planned in a manner that will lead to a meaningful outcome and allow testing the stated hypotheses?

Reviewer #1: Yes

Reviewer #2: Yes

3. Is the methodology feasible and described in sufficient detail to allow the work to be replicable?

Reviewer #1: Yes

Reviewer #2: Yes

4. Have the authors described where all data underlying the findings will be made available when the study is complete?

Reviewer #1: No

Reviewer #2: Yes

5. Is the manuscript presented in an intelligible fashion and written in standard English?

Reviewer #1: Yes

Reviewer #2: Yes

6. Review Comments to the Author

You may also provide optional suggestions and comments to authors that they might find helpful in planning their study.

Reviewer #1: It will be the Editor's decision as to whether or not this description of a study protocol meets the criteria of this journal. I believe the authors have carefully attended to specific recommendations to strengthen the report. Before publication it should be reviewed by a native English-speaker for minor editorial corrections. Note that no data were generated related to this manuscript; the authors have not addressed sharing of future data and so that criterion for publication is also at the discretion of the Editor.

Reviewer #2: Thank you for submitting the revision file. The manuscript is clear to me now so I will accept it for publication.

Thank you

7. PLOS authors have the option to publish the peer review history of their article (what does this mean?). If published, this will include your full peer review and any attached files.

Reviewer #1: No

Reviewer #2: No

---

## [Editor Report · Acceptance letter]

20 Mar 2023

PONE-D-22-33958R1 

Use of “The Knowledge-to-Action Framework” for the implementation of evidence-based nursing in child and family care: study protocol 

Dear Dr. Torres:

I'm pleased to inform you that your manuscript has been deemed suitable for publication in PLOS ONE. Congratulations! Your manuscript is now with our production department. 

Kind regards, 

on behalf of

Dr. Majed Sulaiman Alamri 

Academic Editor

PLOS ONE